# A View on Uterine Leiomyoma Genesis through the Prism of Genetic, Epigenetic and Cellular Heterogeneity [note 1]

**DOI:** 10.3390/ijms24065752

**Published:** 2023-03-17

**Authors:** Alla S. Koltsova, Olga A. Efimova, Anna A. Pendina

**Affiliations:** D.O. Ott Research Institute of Obstetrics, Gynecology and Reproductology, 199034 St. Petersburg, Russia

**Keywords:** uterine leiomyoma, intratumoral heterogeneity, *MED12*, *HMGA2*, chromosomal rearrangements, chromothripsis, epigenetic abnormalities, DNA methylation, histone modifications, non-coding RNAs

## Abstract

Uterine leiomyomas (ULs), frequent benign tumours of the female reproductive tract, are associated with a range of symptoms and significant morbidity. Despite extensive research, there is no consensus on essential points of UL initiation and development. The main reason for this is a pronounced inter- and intratumoral heterogeneity resulting from diverse and complicated mechanisms underlying UL pathobiology. In this review, we comprehensively analyse risk and protective factors for UL development, UL cellular composition, hormonal and paracrine signalling, epigenetic regulation and genetic abnormalities. We conclude the need to carefully update the concept of UL genesis in light of the current data. Staying within the framework of the existing hypotheses, we introduce a possible timeline for UL development and the associated key events—from potential prerequisites to the beginning of UL formation and the onset of driver and passenger changes.

## 1. Introduction

Uterine leiomyoma (UL) is the most common tumour of the female reproductive system, with an incidence rate reaching 70–80% [1]. Despite its benign nature and low risk of malignisation, UL can be characterised by rapid growth, large size and multiple nodes [2]. In approximately 50% of cases, ULs are associated with symptoms that are detrimental to the life quality, including pelvic pain, excessive uterine bleeding and urinary incontinence, and may cause reproductive disorders, in particular, infertility and pregnancy loss [3,4]. 

UL aetiopathogenesis is being actively investigated. Parity, oral contraceptive intake and smoking are among possible protective factors for UL development [5,6,7,8,9]. The risk factors include advanced reproductive age, African and Latin American ethnicity, UL cases in family medical history, obesity, early menarche, arterial hypertension, chronic inflammation, exposure to xenoestrogens in early ontogenesis, sexually transmitted infections and alcohol [8,9,10,11]. Researchers place the highest significance on hereditary predisposition, endocrine and paracrine factors, somatic gene and chromosomal mutations and epigenetic disorders triggered by exposure to endogenous and exogenous factors [9,10,12,13]. Identifying precise mechanisms of UL tumorigenesis and developing universal treatment approaches is complicated by UL being a multifactorial disease with pronounced heterogeneity in clinical, morphological and molecular characteristics. 

The present review analyses the cellular, endocrine, paracrine, genetic and epigenetic aspects of UL tumorigenesis, discusses potential prerequisites and suggests possible key events on the timeline of UL development.

## 2. UL Cellular Origin and Composition

ULs represent dense nodules (myomas) of rounded shape, consisting of transformed cells that produce an excess of extracellular matrix (ECM) [13]. The ECM content varies in different ULs and is presumably subject to change as the myoma progresses and undergoes involution [14]. The main cellular components of a UL are smooth muscle cells, fibroblasts and vascular endothelial cells; other detected cellular components include blood cells, neurons and telocytes [15,16,17,18,19,20,21,22].

Myomas are commonly considered to be monoclonal in nature, with their diverse cellular composition resulting from the expansion and differentiation of a single transformed myometrium cell [9,10,23]. Tissue-specific stem cells are believed to be key to the formation of monoclonal ULs, as they are characterised primarily by their capacity for asymmetric cell division and further differentiation. A cell population with stem/progenitor cell properties (the so-called “side population”) was detected for the first time in the myometrium of non-pregnant women in 2007 [24]. Transcription and cell surface antigen profiles specific to this population were also described [25,26]. Subsequently, a side population expressing several stem markers (*OCT4* (octamer-binding transcription factor 4), *NANOG* (Nanog homeobox), *DNMT3B* (DNA methyltransferase 3B), *GDF3* (growth differentiation factor-3)) was isolated from a UL as well [27,28].

UL stem cells have several differences from those of the myometrium. In culture conditions, ULs form fewer colonies of non-differentiated cells, and the cells, in turn, have a shorter cell cycle [27,29,30]. Stem cells isolated from ULs have also been found to carry *MED12* (mediator complex subunit 12) mutations [29], increased DNA damage and specific expression patterns of genes involved in the repair of single- and double-strand DNA breaks [31]. At the same time, UL stem cells as well as myometrium-derived stem cells are deficient in oestrogen or progesterone receptors [29,32]. 

To grow and proliferate, myometrial and UL stem cells need the presence of differentiated cells with higher levels of hormone receptors. The interaction between stem and mature cells is believed to occur through paracrine signals (WNT (wingless-type) proteins, growth factors, etc.) [28,32,33] or intercellular contacts with telocytes—a type of interstitial cells [17,19,34,35]. Myometrial telocytes express oestrogen and progesterone receptors and can form contacts with various cell types, including smooth muscle, nervous and stem cells, with their thin long extensions called telopodes [36].

The cellular diversity is a considerable, but not the only, contributor to the intratumoral heterogeneity in ULs. Signalling pathways, gene variants, chromosomal abnormalities and epigenetic regulation are among the other aspects of tumorigenesis that create the unique profile of each UL cell (Figure 1).

## 3. Diversity of Endo-, Auto- and Paracrine Mechanisms Regulating UL Development

Sex steroid hormones are the key regulators of UL growth. Not only can UL cells bind oestrogen and progesterone from the blood flow, but they can also convert circulating androgens into oestrogens, which explains higher oestrogen levels in UL tissue as compared to the adjacent myometrium [37].

Testosterone and androstenedione are converted into oestradiol with aromatase and 17β-hydroxysteroid dehydrogenase enzymes. The activity of this enzyme system enables UL cells to produce enough oestrogen to support their growth [38]. Treatment of cultured UL cells with selective aromatase inhibitors abolishes the promoting effect of testosterone and androstenedione on UL cell proliferation [38]. The enzymatic conversion of androgens to oestrogens is also typical of fat tissue [39], making excessive weight a significant risk factor for ULs (Figure 1) [40,41,42,43].

Oestrogens, particularly their most active natural form, 17β-oestradiol, are essential components of many pathways associated with UL development. Oestrogens act through their specific receptors (α and β), which bind to 17β-oestradiol and its natural analogues (estrone and estriol) as well as a variety of xenoestrogens, many of which are considered to be endocrine-disrupting chemicals (EDCs) [44]. Oestrogen receptors bind with oestrogen response elements (EREs) in the genome or with other transcription factors, thus activating the expression of genes coding growth factors, ECM proteins and finally, oestrogen and progesterone receptors [12]. However, UL cell proliferation requires the presence of both oestrogen and progesterone [45,46].

During the luteal phase of the menstrual cycle, characterised by an increased progesterone blood level, a UL features a higher expression of proliferation markers and a higher mitotic activity [45,47,48]. Increased proliferative activity has also been observed in ULs of postmenopausal women following combined oestrogen and progesterone therapy in comparison to those with oestrogen therapy or without hormone replacement [45,49]. Progesterone-mediated UL growth regulation does not only occur through increased proliferation but also through decreased apoptosis and higher ECM accumulation [46,50,51]. Directly or via interacting with other transcription factors, progesterone receptors can regulate the expression of many genes, in particular, proliferating cell nuclear antigen (PCNA), epidermal growth factor (EGF), transforming growth factor beta (TGF-β), anti-apoptotic Bcl-2 protein and the miR-29b microRNA, which regulates the accumulation of ECM [37,52]. 

Oestrogen and progesterone also have non-genomic mechanisms of action in UL cells [53,54], realised through the binding of activated hormone receptors with cytoplasmic and membrane regulatory proteins involved in various kinase cascades—MAPK (mitogen-activated protein kinase), PI3K/AKT (phosphatidylinositol 3-kinase / protein kinase B), PLC/PKC (phospholipase C / protein kinase C) and cAMP/PKA (cyclic AMP / protein kinase A) [54,55]. Paracrine signalling pathways also facilitate intercellular communication in myometrial and UL tissues [24,34]. UL-specific *MED12* mutations are linked, among other factors, to changes in the expression and functioning of various WNT/β-catenin signalling components [56,57], whereas the inhibitors of the WNT/β-catenin pathway (vitamin D, Simvastatin, etc.) can suppress UL cell proliferation and ECM accumulation by lowering the expression of oestrogen and progesterone receptors [58,59,60,61].

Non-hormonal UL development regulators are growth factors (fibroblast growth factor (FGF), vascular endothelial growth factor (VEGF), epidermal growth factor (EGF), insulin-like growth factor (IGF), platelet-derived growth factor (PDGF), transforming growth factor β (TGF-β)), proinflammatory cytokines (tumour necrosis factor TNF-α) and ECM components [62]. The UL ECM includes collagens, fibronectin, laminins and proteoglycans, which, in turn, facilitate mechanotransduction and the launch of signalling pathways (WNT/β-catenin, ERK (extracellular signal-regulated kinase)/MAPK and Hippo/YAP (yes-associated protein) pathways) [63,64]. Therefore, the ECM is not just a side product of the UL cell anomalous functioning but an active participant in the neoplastic process thanks to its reciprocal interaction with UL cells.

Thus, oestrogen and progesterone synergistically regulate the mechanisms of cell proliferation, differentiation and fibrosis in the UL through their receptors. The UL’s overlapping network of intra- and intercellular signalling goes beyond sex hormones to include a variety of growth factors, receptors, cellular adhesion molecules, cytokines, ECM proteins and kinase cascades associated with them. Not only can the reprogramming of fundamental hormonal, paracrine and immune response mechanisms result from genetic and epigenetic changes, but it can serve as an independent factor triggering genetic instability and aberrant gene expression in UL cells.

## 4. The Role of Woman’s Genotype in UL Aetiology

Researchers have theorised about the hereditary nature of ULs for a rather long time, considering the ample evidence of a higher UL frequency among close blood relatives. In the 1950s, *The Lancet* published medical data on four generations of one family, including 15 women over 15 years old [65]. According to the family history, nine of the women were reliably found to have ULs, which attested to the high significance of hereditary factors in the development of ULs in the female members of this family. A later study, which included 215 women from 97 families, revealed that the risk of UL in women whose mothers had ULs almost doubled [66]. The role of heredity in UL aetiology was confirmed in the study of mono- and dizygotic twins [67] and several ethnic groups [68,69]. 

Women of ethnic Latin American and African ancestry have a considerably higher risk of developing ULs than women of European ancestry [1]. Furthermore, African ancestry is additionally associated with earlier development of ULs and heavier symptoms [70]. Along with hereditary factors, vitamin D deficiency, which often occurs in black women, dietary specifics (a high share of soy products, in particular), obesity, high-stress levels and a high content of stem cells in the myometrium are also considered to be of significance for the observed differences [9].

Modern genomic research (genome-wide association search (GWAS) and genome-wide single-nucleotide polymorphism analysis) has identified variants determining predisposition for UL in over 50 genes (*WNT4* (Wnt family member 4), *GREB1* (growth regulating oestrogen receptor binding 1), *TERC* (telomerase RNA component), *TERT* (telomerase reverse transcriptase), *HMGA1* (high mobility group AT-hook 1), *FOXO1* (forkhead box O1), *TP53* (tumour protein p53), etc.) [71,72,73,74,75,76,77,78,79,80,81]. These genes are involved in DNA damage repair, telomere length maintenance, hormonal and paracrine regulation, apoptosis, urogenital system development and early menarche. Furthermore, so-called “protective” single-nucleotide variants, which reduce UL risk, have been discovered in the introns of the following genes: *KCNMB2* (potassium calcium-activated channel subfamily M regulatory beta subunit 2), *FBN2* (fibrillin 2), *ESR1* (oestrogen receptor 1) and *CELF4* (CUGBP Elav-like family member 4) [82]. Researchers have also described *COL6A3* (collagen type VI alpha 3 chain), *COL13A* (collagen type XIII alpha 1 chain), *ARHGAP26* (Rho GTPase activating protein 26), *MAN1C1* (mannosidase alpha class 1C member 1), *BET1L* (Bet1 golgi vesicular membrane trafficking protein like), *TNRC6B* (trinucleotide repeat containing adaptor 6B) and *COMT* (catechol-O-methyltransferase) gene variants associated with the accumulation of ECM in the UL tissue, the size, localisation and the multiple form of UL [83,84,85,86,87].

Particular hereditary syndromes (Reed’s, Alport, Proteus and Cowden syndromes), which feature a high risk of UL [10], warrant special attention. Frequently referred to as the HLRCC syndrome (Hereditary Leiomyomatosis and Renal Cell Carcinoma), Reed’s syndrome is caused by germline mutations in *FH*, the gene encoding fumarate hydratase, a tricarboxylic acid cycle enzyme [88,89]. Patients with this syndrome face a high risk of developing renal cell carcinoma and benign leiomyomatosis, with women displaying a high frequency of multiple ULs. With Reed’s syndrome, the neoplastic process is triggered through biallelic *FH* gene inactivation, fumarate accumulation in the cells, changes in cell metabolism, the stabilisation of hypoxia-inducible factors (HIFs), the activation of carcinogenic transcription factor NRF2 (nuclear factor erythroid 2 related factor 2) and changes to the expression profile [9,10]. In most cases, however, the development of FH-deficient ULs is associated with somatic but not germline *FH* mutations (whole-gene deletions, frameshift or missense mutations) [90,91,92,93]. ULs with *FH* mutations are often histologically distinctive, demonstrate hypercellularity, nuclear atypia, inclusion-like nucleoli, stromal oedema, and occur with a high frequency among atypical leiomyomas (37.3% compared to 1.6% in unselected ULs) [91,92,94]. 

Alport syndrome is caused by basement membrane damage resulting from germinal mutations in the genes of type IV collagen (*COL4A3*, *COL4A4*, *COL4A5* (collagen type IV alpha 3, 4, 5 chain genes)). Patients with this syndrome experience primarily progressive loss of kidney function and hearing loss [95]. ULs are considerably less frequent in patients with Alport syndrome than with Reed’s syndrome, and their development is associated with mutations at the *COL4A5-COL4A6* locus. Overall, the contribution of *FH* and *COL4A5-COL4A6* mutations in the total UL frequency is low, not exceeding 5% [96,97].

Therefore, the significance of heredity in UL aetiology has been confirmed with extensive data. At present, however, genotype alone is not sufficient for an accurate prognosis of UL development in a woman except in cases of rare hereditary syndromes. ULs are associated with allelic variations in genes involved in a wide range of processes, which presumably determines its high occurrence rate and further emphasises its multifactorial nature (Figure 1).

## 5. Spectrum of Somatic Genetic Aberrations in UL Cells

ULs are characterised by varied chromosomal abnormalities and high heterogeneity in acquired somatic mutations, with the most frequent being harboured in *MED12, HMGA2* (high-mobility group AT-hook 2), *FH* and *COL4A5-COL4A6* genes. Less frequent mutations in *MED8* (mediator complex subunit 8), *MED18* (mediator complex subunit 18), *CDK8* (cyclin-dependent kinase 8), *CASC15* (cancer susceptibility 15), *COL4A6*, *DCN* (decorin), *AHR* (aryl hydrocarbon receptor), *NRG1* (neuregulin 1), *ADAM18* (ADAM metallopeptidase domain 18), *HUWE1* (HECT, UBA and WWE domain containing E3 ubiquitin protein ligase 1), *FBXW4* (F-box and WD repeat domain containing 4), *FBXL13* (F-box and leucine-rich repeat protein 13) and *CAPRIN1* (cell cycle-associated protein 1) genes [98] and mitochondrial DNA mutations are also associated with ULs [99].

### 5.1. MED12 Mutations 

In different samples, the frequency of ULs with *MED12* mutations (MED12-ULs) varies from 31% to 85% [100,101,102,103,104,105]. The presence of MED12-ULs in women is associated with a history of inflammatory disorders of the pelvic organs, nulliparity and African ancestry [105,106]. *MED12* mutations are detected more frequently in multiple ULs and small, mitotically active myomas of normal morphology and subserous (as opposed to intramural) localisation with a high ECM content [102,103,106,107,108,109,110,111]. Furthermore, a UL susceptibility locus was identified 250 kb upstream of the *MED12* gene [75]. However, the environment plays an important role in the emergence of *MED12* mutations because women with multiple ULs can have *MED12* mutations in some of the myomas but not all of them.

The *MED12* gene is located in the Xq13.1 chromosome region and encodes the large mediator complex subunit (the so-called “Mediator”), which bridges gene-specific regulatory proteins in the promoter region to the RNA polymerase II initiation complex. Pathological variants in *MED12* are represented in ULs primarily by heterozygous missense mutations in codon 44 of exon 2; ULs have also been found to feature missense mutations in codons 36 and 43, in-frame deletion plus insertion mutations and in rare cases, exon 1 mutations [100,112]. *MED12* mutations have been detected in UL stem cells [29] but not in the adjacent myometrium or the UL pseudocapsule [113] and are therefore treated as the potential drivers of UL tumorigenesis.

In MED12-ULs, *MED12* is expressed primarily from the mutant allele [100], indicating the involvement of the anomalous variant of MED12 protein in the neoplastic process. A UL mouse model demonstrates that the expression of *MED12* c.131G>A mutant allele in uterine tissue is sufficient for the development of UL-like tumours with chromosomal aberrations [114]. However, the precise molecular mechanism of how *MED12* mutations trigger UL growth remains to be established. *MED12* mutations undermine the transduction of several cytoplasmic and transcription signals due to the disrupted ability of the Mediator to activate cyclin C and CDK8/CDK19 cyclin-dependent kinases [115,116,117]. Furthermore, *MED12* mutations are associated with an increased expression of WNT/β-catenin signalling and changes in the regulation of downstream pathways (mTOR, TGF-β) linked to cell proliferation, autophagy and ECM accumulation [56,57,118]. Other signalling pathways—MAPK, AKT and IGF (insulin-like growth factor) —mediate the action of oestrogen and progesterone in MED12-ULs. Anomalous MED12 protein interacts with the progesterone receptor more efficiently, driving the progesterone-dependent expression of potential carcinogens in MED12-ULs (in particular, this has been shown for the receptor activator of nuclear factor kappa-Β ligand (*RANKL*) and tryptophan 2,3-dioxygenase (*TDO2*) genes) [119,120]. Gene expression in MED12-ULs is additionally affected by epigenetic abnormalities including post-translation modifications of histones, enhancer architecture and DNA methylation [111,121]. Nevertheless, changes in the global level of UL DNA hydroxymethylation are not associated with *MED12* mutations [122]. Smooth muscle cells from MED12-ULs also exhibit signs of replication stress, such as higher levels of R-loops, altered replication fork dynamics and delayed S phase of the cell cycle [123].

Importantly, not all cells of a MED12-UL carry *MED12* mutations. Within the first several passages in culture, MED12-ULs manifest an elimination of cells with *MED12* mutations with the expansion of wild-type cells [124]. Wu et al. have established that *MED12* mutations are present in smooth muscle cells but not tumour-associated fibroblasts [109]. However, Goad et al. have recently used single-cell RNA sequencing to show that intercellular heterogeneity of *MED12* mutations is typical for all three predominant UL populations: fibroblasts, smooth muscle cells and endothelial cells [22]. Variations in the mechanisms of in vitro maintenance of heterogeneous MED12-UL cell populations could include the differential effect of sex steroids because UL smooth muscle cell proliferation is stimulated mostly by progesterone, while that of fibroblasts is oestrogen-regulated [109,125]. In light of this, it is practical to prioritise the optimisation of culture conditions, mutant UL cell immortalisation and use of tumour xenografts when creating cellular and organotypic MED12-UL models [57,126,127,128,129].

*MED12* mutations could either be the only genetic aberration detected in a myoma [118,130,131] or coexist with other “driver” mutations (*HMGA2*, *FH*, *COL4A5*-*COL4A6*) [132,133]. Furthermore, karyotypically normal and abnormal ULs feature the same frequency of *MED12* mutations [101,134], which do not affect the growth potential of cells with chromosomal aberrations in vitro [124,134,135]. 

### 5.2. Chromosomal Abnormalities 

At least 40% of ULs carry chromosomal abnormalities. Initially, they were detected using karyotyping of short-term and long-term tumour cell cultures. Over the last few decades, cytogeneticists have described over 200 UL-specific abnormalities, of which the most frequent are structural rearrangements involving chromosome regions 1p, 6p21, 7q and 12q15 [136]. In a significant share of ULs, chromosomal abnormalities were presented in the mosaic state alongside karyotypically normal cells, which invited hypotheses of their secondary origin within the tumour or in culture conditions [137,138,139]. The introduction of FISH analysis on interphase nuclei ushered in the discovery that mosaic chromosomal rearrangements may be present in uncultured myomas and presumably emerge in vivo [140]. The further advancement of molecular technologies opened opportunities for the analysis of chromosomal aberrations with the use of genomic DNA from the native tumour. Researchers identified the genes involved in the most frequent rearrangements, chromosomal abnormalities in cell clones incapable of in vitro growth and extremely complicated variants of chromosomal aberrations, such as chromothripsis [134,141,142,143]. In our opinion, conventional and molecular techniques are complementary rather than competitive and create a more objective picture for a better understanding of the cytogenetic aspects of UL pathogenesis.

#### 5.2.1. 12q14-15 Rearrangements

Region 12q14-15 was among the first to be referred to as a “hot spot” of chromosomal rearrangements in ULs [144]. This region is characterised primarily by reciprocal changes involving region 14q24, with fewer frequent translocations involving two or more other chromosomes and intrachromosomal aberrations [136]. The development of ULs carrying 12q14-15 rearrangements has been associated with the *HMGA2/HMGI-C* locus [141,145,146]. The *HMGA2* gene encodes a non-histone protein that regulates chromatin conformation and gene expression via binding with AT-rich DNA sequences of enhancers and promoters. Normally, this gene is active in embryonic tissues, stem cells and many malignant tumours [147,148,149]. 

In ULs, most 12q14-15 breaks are located upstream of the promoter sequence of the *HMGA2* gene, upregulating its expression [146,150,151]. Along with transcriptional alterations, the higher levels of HMGA2 protein in ULs could be associated with the loss of *let-7* regulatory microRNA binding sites in truncated or chimeric transcripts of the damaged *HMGA2* gene [152,153]. Furthermore, the overexpression of *HMGA2* is typical for a considerable subset of ULs without 12q14-15 rearrangements [154] and could be caused by either *HMGA2* hypomethylation or decreased levels of repressor microRNAs: *let-7a*, *miR-26a*, *miR-26b*, *miR-93* and *miR-106b* [133,155]. Therefore, regardless of the presence or absence of 12q15 rearrangements, all ULs featuring *HMGA2* overexpression are consolidated into a single pathogenetic subgroup, HMGA2-ULs.

HMGA2-ULs contain 90% of *HMGA2*-positive smooth muscle cells and are characterised by a larger size and higher growth rate, mostly through mitotic activity rather than excessive ECM accumulation [108,109,145]. HMGA2-ULs respond to gonadotropin-releasing hormone agonist therapy less than karyotypically normal ULs, which suggests a lower dependence of HMGA2-ULs on circulating sex steroids [156]. In standard culture conditions without hormone supplementation, the cells of a UL with rearrangements at the *HMGA2* locus are stably maintained throughout multiple passages, but their growth is coupled with a decrease in proliferation level and *HMGA2* expression and a parallel increase in cellular senescence markers [124,157,158,159].

It has been demonstrated in an animal UL model that the expression of a shortened *HMGA2* variant in myometrial stem cells is sufficient for their transformation into progenitor UL cells [160]. HMGA2-ULs are further characterised by the activation of AKT and IGF signalling pathways, increased angiogenesis and the overexpression of the oestrogen receptor α (*ESR1*), the *PLAG1* (pleomorphic adenoma gene 1) proto-oncogene, *CCND1*, *CCND2*, *CCND3* and *CDK6* cell cycle regulators, epidermal (*EGF*) and transforming (*TGFA*) growth factors, the fibroblast growth factor (*FGF2*) and the vascular endothelial growth factor (*VEGFA*) [20,118,161,162,163,164,165]. Furthermore, the functioning of genes located on the other chromosomes involved in the translocations with the 12q14-15 region may be also affected. Described cases include the genes of a DNA repair enzyme *RAD51B*/*RAD51L1* (14q24), G2/M cell cycle checkpoint regulator *CCNB1IP1*/*HEI10* (14q11), cytochrome C oxidase subunit *COX6C* (8q22), cytokine response regulator *TRAF3IP2* (6q21) and Golgi complex proteins *NAA11* (4q21) and *COG5* (7q22-31) [166,167,168,169,170,171,172]. At the same time, the rapid progression of a tumour, and, therefore, a possible manifestation of genetic instability is hampered in HMGA2-ULs by the activity of pathways associated with autophagy and cellular senescence [158,165].

#### 5.2.2. 6p21 Rearrangements

As another distinctive trait of ULs, rearrangements in chromosome region 6p21 are associated with the damage and overexpression of the *HMGA1*/*HMGI-C* gene, which emphasises the contribution of the HMGA protein family to tumorigenesis [173,174]. Similar to *HMGA2*, increased *HMGA1* expression is mediated by the hypomethylation of CpG islands at the promoter gene region and may present in a UL without 6p21 rearrangements [155,175,176]. HMGA1-ULs are characterised by the activation of the *PLAG1* proto-oncogene and *PAPPA2* (pappalysine), the insulin-like growth factor bioavailability regulator, and inhibited expression of *SMOC2*, a gene involved in angiogenesis [118,155]. 

#### 5.2.3. 10q22 Rearrangements

Apart from locus 14q24 (the *RAD51B* gene), chromosome region 10q22 is another frequent translocation partner for regions 12q14-15 and 6p21 [136,177]. The rearrangements of this region in ULs affect the histone acetyltransferase gene, *KAT6B*/*MORF*/*MYST4* (lysine acetyltransferase 6B), which reflects the involvement of anomalous chromatin regulation in UL development [178]. Whereas the range of *KAT6B* abnormalities and their impact on clinical and morphological UL characteristics have barely been studied, there are data suggesting that chimeric transcripts of *KAT6B*-*KANSL1* (17q21) may be discovered in fast-growing cellular ULs, although they are most typical for malignant uterine tumours [179,180].

#### 5.2.4. Deletions in the Long Arm of Chromosome 7

Detected in 17–50% of karyotypically abnormal ULs, deletions in the long arm of chromosome 7 (7q) are also among the most frequent chromosomal rearrangements in ULs [136,138,181]. In culture conditions, UL cells with a deletion in 7q undergo negative selection revealing their coexistence with the 46,XX cells within the first few passages, enabling researchers to suggest the secondary nature of this abnormality early on [139,181]. The significance of deletions in 7q for the UL karyotype evolution is confirmed by their recurrent emergence as independent clonal events or secondary emergence in cells with pre-existing chromosome abnormalities [134,136,182,183]. There is also a documented case of a UL with deletions in the long arms of both chromosome 7 homologues [184].

While ULs have been found to feature deletions along the entire long arm of chromosome 7, they most frequently affect region 7q22 [185,186,187,188,189]. This region is characterised by an extremely high gene density, considerably complicating the search for mechanisms associated with UL pathogenesis. The genes located in 7q22 are involved in a wide range of regulatory processes, including collagen production, various signalling pathways, and DNA repair [183]. The molecular studies of ULs harbouring deletions, translocations and inversions involving region 7q22 have demonstrated losses and damage in gene *CUX1*/*CUTL1* (cut-like homeobox 1), which could serve as a haploinsufficient tumour suppressor [118,190,191]. Deletions in the origin recognition complex protein gene *ORC5*/*ORC5L* and the transmembrane protein gene *LHFPL3* are also presumed to be of significance [192,193,194]. Furthermore, deletions in 7q are associated with changes in the expression of several genes located in the region: *LMTK2* (lemur tyrosine kinase 2), *COPS6* (COP9 signalosome subunit 6), *CUX1*, *MLL5*/*KMT2E* (lysine methyltransferase 2E), *LHFPL3* (LHFPL tetraspan subfamily member 3), *LAMB1* (laminin subunit beta 1) and more [118,194,195,196].

An important characteristic of ULs with deletions in 7q (7q-ULs) is their similarity to karyotypically normal ULs in terms of size and response to gonadotropin-releasing hormone agonist therapy (as opposed to ULs with rearrangements in 12q14-15 region) [197]. The mosaic deletions in 7q and the lack of unique clinical and morphological traits in this group of ULs attest to the high possibility of these rearrangements occurring as “passenger” genetic aberrations, without any impact on the UL growth potential. The possible exception is cases of secondary deletions in 7q in cells with pre-existing chromosome abnormalities. Thus, the emergence of a deletion in 7q in cells with a t(12;14) translocation offers them a proliferative advantage both in vivo and in vitro, not only on karyotypically normal cells but also on the initial population of cells carrying the translocation [134].

#### 5.2.5. Chromosome 1 Rearrangements

Various chromosome 1 rearrangements, which may occur in both karyotypically normal and karyotypically abnormal cells, also significantly contribute to the clonal UL evolution. ULs are characterised by the formation of ring chromosomes 1 and translocations involving both homologues of chromosome 1 or other chromosomes [134,198,199,200,201]. Chromosome 1 rearrangements in ULs are often accompanied by deletions [202]. Partial monosomy 1p occurs more frequently in large cellular ULs—a rare histological UL subtype presumably characterised by a higher risk of malignisation [203,204].

#### 5.2.6. Chromothripsis

Chromothripsis is a special case of complex chromosomal rearrangements characterised by multiple breaks and deletions in one or few chromosomes [205]. Initially, chromothripsis was discovered using mate-pair sequencing in malignant tumours and was associated with high-grade cancer and a poor prognosis [206,207]. However, in 2013, Mehine et al. documented the first case of chromothripsis in UL cells [142]. Based on various estimates, this variant of complex chromosomal rearrangements occurs in 20–41.7% of ULs [97,142,143]. 

Compared to malignant tumours, chromothripsis in UL cells is characterised by a lower number of breaks (around twenty or more versus tens to hundreds) and a larger number of chromosomes involved (up to four) [208]. Chromothripsis has been documented in myomas that show no signs of malignancy and have no *MED12* or *FH* mutations. Furthermore, the use of molecular techniques allows for finding chromothripsis in ULs with a previously detected apparently normal karyotype or other chromosomal abnormalities indicating the decrease in the proliferative potential of cells with chromothripsis in vitro [134,143]. Meanwhile, there is a documented case of unbalanced chromothripsis, detected both in cultured and uncultured UL cells [209]. Most likely, the proliferative capacity of UL cells with chromothripsis in culture conditions is determined to a lesser extent by the complexity of chromosomal aberrations and to a greater extent by the genome regions involved in the rearrangement. 

#### 5.2.7. Other Chromosomal Abnormalities 

Regions involved in chromosomal rearrangements in ULs are located almost throughout the entire genome. Apart from the chromosomes and chromosomal regions described in this section, research papers on UL cytogenetics often mention rearrangements in regions 2p, 2q, 3p, 3q, 5q, 13q, 19q, 22q, Xp, trisomy 12 and monosomy 10 and 22 [134,136,210].

### 5.3. The Clinical Significance of Somatic Genetic Abnormalities in UL Cells

The high diversity of somatic changes in ULs is meaningful not only for fundamental studies of UL pathogenesis but also in the development of treatment approaches. Unlike malignant neoplasms, the correlation between the UL response to pharmacological therapy and its genotype/karyotype has been barely examined. For the moment, ULs with a translocation t(12;14)(q14-15;q24) have been established to show less shrinkage in response to gonadotropin-releasing hormone agonist therapy than ULs with deletions in 7q or with a normal karyotype [156,197]. The same is true for ULs with *MED12* mutations as compared to the “wild-type” ULs [211]. However, MED12-ULs have 4.4 times higher odds of shrinking in response to treatment with the progesterone receptor modulator ulipristal acetate as compared to HMGA2-ULs [212].

Data on genetic aberrations also serve as an auxiliary tool in the clonal origin analysis of ULs and other tumours. Whereas the majority of multiple myomas in the uterus emerge independently from one another, in some cases, different myomas could be clonally linked and could share the same chromosomal abnormalities [139,213,214,215,216]. Molecular studies have shown that such ULs can additionally acquire individual genetic mutations conformed with the branched tumour evolution model [217]. Furthermore, in extremely rare cases, the development of benign tumours clonally linked to the initial myoma is observed not only in the corpus of the uterus but also in remote tissues, primarily in the lungs [218]. Some studies also confirmed shared histological, genetic and cytogenetic profiles between secondary tumours and the primary myoma [219,220,221,222,223,224,225]. Deletions in the long arms of chromosomes 3, 11, 19 and 22 [222,226], *FH* mutations [227] and the activation of the alternative telomere lengthening mechanism could be associated with the risk of benign UL metastasising [228]. Pathogenesis of benign metastasising (or parasitic) ULs is barely studied. There are assumptions that UL metastases might occur as a complication after laparoscopic surgery due to the morcellation of myomas and even that UL metastasising points towards the underestimated malignant potential of the initial tumour [229,230,231,232]. Regardless of UL metastasising mechanism, this phenomenon raises the need to update UL classification criteria as well as criteria to distinguish benign and malignant tumours.

Areas with UL-specific histology may be present in uterine leiomyosarcomas, attesting to the possibility of UL malignisation [233,234]. Uterine leiomyomas and leiomyosarcomas often feature a similar spectrum of genetic alterations, although leiomyosarcoma undoubtedly demonstrates a more pronounced cytogenetic instability [202,235,236,237]. The fact that warrants special attention is that, whereas deletions in chromosomes 1 and 22 are associated with UL malignisation risk [231,238], chromothripsis is not [142,143,209]. Overall, UL malignisation is a considerably rare event and occurs more typically in histopathological variants [239].

Karyotype abnormalities in UL cells could also lead to false positive results in non-invasive prenatal testing (NIPT), a modern technique for detecting foetal chromosomal abnormalities based on the analysis of circulating cell-free foetal DNA in the mother’s blood [240,241,242]. Pregnant women with ULs run a higher risk of having unbalanced subchromosomal aberrations detected through NIPT—but not trisomy 13, 18 or 21 or sex chromosome aneuploidies. In this category of patients, the most frequently detected abnormality is deletions in 7q, which is determined by their frequent occurrence in ULs [243].

Therefore, the genetic alterations should be taken into account not only in the investigation of UL tumorigenesis mechanisms but also when assessing the risk of its benign metastasising and malignisation, developing efficient treatment approaches and diagnosing other diseases and conditions.

### 5.4. Speculations on the Origin of Somatic Genetic Abnormalities in UL Cells

The occurrence mechanisms of genetic abnormalities specific to ULs have been investigated in parallel with the discussion about the clonal origin of myomas. As early as in the 1960s, researchers analysed the expressed alleles of glucose-6-phosphate dehydrogenase and established the non-random X chromosome inactivation in myomas, which attested to possible UL monoclonality [244]. The discovery of mosaic chromosomal abnormalities in some ULs brought new relevance to the question of their clonal origin, with the answer provided before long by the inquiry into the androgen receptor gene *AR* methylation status, which confirmed the non-random inactivation of chromosome X in this group of ULs as well [139]. Since then, the monoclonal origin of ULs has been acknowledged by the vast majority of authors. The accumulation of data confirming the unstable repeatability and discordance of *AR* methylation analysis results obtained using varied techniques cast a shadow of doubt on the matter [245]. Subsequently, Holdsworth-Carson et al. used the analysis of *AR* expression (RNA-HUMARA) instead of *AR* methylation to confirm the monoclonal origin in 23/25 (92%) examined ULs, including for individual tumour cell populations: fibroblasts, smooth muscle cells and endothelial cells [23]. However, the molecular techniques referred to above allow for the analysis of multicellular samples but not individual cells. This may result in misinterpretation of a single cell clone overgrowth as monoclonality and thus the underestimation of minor clones in ULs. Such methodological limitations present an immense challenge for investigating a chain of events that have been driving the UL formation and growth up to the time of analysis. 

Since the discovery of *MED12* mutations in UL stem cells [29], many researchers began to refer to this genetic aberration as a “driver” of the condition, while actively discussing the potential mechanisms of *MED12*-associated myometrial cell transformation [13,106,118,246]. However, the analysis of individual MED12-UL populations demonstrated their genetic heterogeneity, providing a new incentive for further investigation of the clonal UL origin [22,109]. MED12-ULs are presumed to be non-monoclonal and capable of recruiting surrounding wild-type cells to support tumour growth [22]. While acknowledging the validity of this hypothesis, we would nevertheless like to highlight the possibility of genetic heterogeneity within the MED12-UL stem cell population, and, therefore, the emergence of *MED12* mutation not as a transforming event but in the early stages of tumorigenesis (Figure 2). 

As for chromosomal abnormalities, the consensus about their secondary occurrence during tumour growth has held until the present day. The exact timing and mechanisms, however, remain to be established. Not all ULs are cytogenetically stable, with a share of myomas characterised by karyotype evolution [134,136,200]. Minor populations of heteroploid cells have also recently been detected in ULs with an apparently normal karyotype [135], demonstrating that the risk of chromosomal abnormalities persists for ULs in the long-term perspective.

Rearrangements involving region 12q14-15 (primarily balanced translocations and inversions) are presumed to occur in the early stages of UL formation, while deletions in chromosomes 1 and 7 occur in later stages [181,199,200,201]. The start of UL formation is most likely accompanied by global epigenetic changes in stem cells. Actively transcribed genes located on different chromosomes may become topologically closer to each other in the transcription factories, which, in turn, may result in faulty repair of multiple DNA breaks and the emergence of structural rearrangements [247,248]. Data on *HMGA1* (6p21) and *HMGA2* (12q14-15) overexpression in some ULs without the rearrangements of corresponding chromosome regions [154,175,176], a similar level of *HMGA2* expression across multiple myomas in one patient, the localised pattern of its expression in the myometrium of UL patients [132] and the differential gene expression profile in the myometrium of patients with *HMGA2*- and 7q-ULs speak in favour of this suggestion [162].

Since UL cell proliferation is not coupled with telomerase or ALT (alternative telomere lengthening mechanism) activity, ULs are characterised by shorter telomeres than the myometrium [134,228,249,250,251,252]. Telomere attrition is directly linked to the complex rearrangements (including chromothripsis) and may trigger chromosomal instability in the later stages of UL growth [134,253,254]. Alternatively, the impact of mechanical forces, similar to those occurring during pregnancy, may also serve as a factor triggering cytogenetic aberrations in the cells of developing ULs [255].

Therefore, we are suggesting that, in the period from the beginning of tumour mass formation to its surgical removal, a UL experiences multiple changes in dominating processes that may cause somatic mutations. In the early stages, the more significant factors are a hereditary predisposition, chromatin-level changes, overexpression of individual genes and abnormal hormonal regulation, whereas in the late stages, they give way to telomere shortening, deterioration in DNA repair systems, general metabolic and functional abnormalities in UL cells, the mechanical influence of ECM and adjacent tissues and finally, hormonal and physiological changes in the female body associated with pregnancy, childbirth, lactation and the late reproductive period. Nevertheless, it remains unclear, considering the intercellular genetic heterogeneity of MED12-ULs, whether cytogenetic changes occur in wild-type cells or cells carrying *MED12* mutations and how acquired chromosomal abnormalities can change the “developmental programme” of MED12-ULs.

## 6. Epigenetic Changes Associated with UL Development

An epigenome is a record of various chemical markers to the DNA and histone proteins and molecules involved in gene expression regulation on the pre- and post-transcription levels. The functional role of epigenetic factors goes beyond genome regulation to include changes in chromosome compaction throughout the cell cycle, centromere positioning, ensuring DNA availability for its recognition by repair complexes, intercellular communication and more [208,256]. Errors in epigenetic processes have been described for multiple conditions, including the UL. The aberrant expression of multiple genes in UL cells results from epigenetic changes on all levels, from DNA methylation and histone modifications to non-coding RNAs [257].

### 6.1. Histone Post-Translational Modifications in ULs

Histones (H1, H2A, H2B, H3, H4) are highly conserved DNA-binding proteins whose primary functions are packaging DNA into nucleosomes and chromatin compaction. Not only do the post-translational modifications (acetylation, methylation, phosphorylation, SUMOylation, ubiquitination, etc.) of amino-acid residues in N- and C-terminal regions of histone proteins ensure the dynamic structure of chromatin, but they are also involved in gene expression regulation, the interaction of DNA with other proteins and the processes of chromosome replication, repair, recombination and disjunction [258]. Acetylation of lysine residues in histones H3 and H4 is a marker of transcriptionally active (open) chromatin. Mono-, di- and trimethylation of lysine residues in histone H3 and H4 can indicate either active (H3K4, H3K36, H3K79) or repressed chromatin (H3K9, H3K27, H4K20). Histone H3 phosphorylation at serine and threonine residues also contributes to chromatin structure regulation and plays an important part in the processes of kinetochore function, mitotic chromosome condensation/segregation and centromere recognition by specific proteins. Furthermore, cross-interaction among different types of histone modifications may be observed [259]. Exploration of the spectrum and functional significance of histone post-translational modifications is a rapidly expanding area of epigenetic research that has gained particular relevance in the context of developmental biology and tumorigenesis.

ULs have been found to feature various changes in histone modification patterns and associated enzymatic systems. ULs are characterised by increased histone deacetylase expression levels as compared to the myometrium [260,261]. In UL cells, histone deacetylase 6 is a positive regulator of oestrogen receptor α gene expression [260]. If a UL is exposed to oestrogen or a combination of oestrogen, progesterone and mifepristone, its histone deacetylase activity reaches higher levels than in the myometrium [262]. Histone deacetylase inhibition in UL cells results in a dose-dependent decrease in the expression of β-catenin, cyclin D1, c-Myc, inhibited growth and proliferation, apoptosis induction, cell cycle arrest and decreased TGF-β3 signalling and ECM production [261,263].

Furthermore, around 70% of ULs feature higher expression of histone methyltransferase *EZH2* (the enhancer of zeste homologue 2) and content of H3K27me3 histone than the myometrium [264]. This enzyme is a negative regulator of DNA repair gene activity in ULs. Exposure of UL cells to azacitidine and other EZH2 inhibitors drives the expression of *RAD51B*, *BRCA1* (BRCA1 DNA repair associated) and *MSH2* (mutS homolog 2) genes [264,265]. In Eker rats (the animal UL model), neonatal exposure to phytoestrogen genistein decreases EZH2 activity and H3K27me3 content in the uterine tissue, which results in an increased UL incidence and tumour multiplicity later in life [266]. Administration of genistein and cadmium, a “metalloestrogen”, to UL cells is also associated with increased content of H3S10ph histone and higher proliferation level [267,268].

A recent study has documented constitutional variations in the genes *YEATS4* and *ZNHIT1* encoding chromatin remodelling complex proteins (SRCAP), which predispose to UL development [269]. Somatic mutations of these and other genes encoding SRCAP proteins in ULs result in the loss of histone H2A.Z, an active chromatin marker, and changes in the tumour’s expression profile [269].

### 6.2. DNA Methylation and Demethylation in ULs 

Another gene expression regulation mechanism is the methylation of cytosine at the fifth position (DNA methylation). For example, 5-methylcytosine blocks the binding of transcriptional factors with DNA directly or indirectly through specific protein complexes [270]. The transfer of the CH_3_ group from S-adenosylmethionine to cytosine is catalysed by DNA methyltransferases. DNA methyltransferase 1 (DNMT1) maintains the methylated state of the DNA in dividing cells by attaching the methyl group to cytosine residues in the newly replicated strand using the parental strand as a template. DNA methyltransferases 3A and 3B (DNMT3A, DNMT3B) catalyse methylation de novo and can methylate cytosine both in hemimethylated and unmethylated DNA [271]. The 5-methylcytosine content in a cell is also determined by the TET (Ten-Eleven Translocation) oxygenase enzymes, the AID/APOBEC deaminating complex (activation-induced deaminase/apolipoprotein B mRNA editing enzyme), TDG (thymine DNA-glycosylase) glycosylases and SMUG1 (selective monofunctional uracil DNA glycosylase), which are involved in the reverse process: DNA demethylation [272].

TET enzymes catalyse the sequential oxidation of 5-methylcytosine to 5-hydroxymethylcytosine, 5-formylcytosine and 5-carboxylcytosine [273,274,275]. The activity of TET oxygenases requires ascorbic acid, Fe(II) and α-ketoglutarate [273]. The inhibition of TET-dependent DNA demethylation may result from either a deficiency of cofactors or the accumulation of 2-hydroxyglutarate, succinate and fumarate in the cell due to a metabolic disorder or mutations (for example, in the fumarate hydratase gene *FH*) [276,277]. Decreased 5-hydroxymethylcytosine content may also occur as a result of adverse environmental exposure [278]. The AID/APOBEC complex can deaminate 5-methylcytosine to thymine and 5-hydroxymethylcytosine to 5-hydroxymethyluracil [279]. The excision repair of oxygenated and deaminated products of 5-methylcytosine is performed by TDG and SMUG1 DNA glycosylases. SMUG1 converts 5-hydroxymethyluracil to cytosine, whereas TDG catalyses the production of cytosine from thymine, 5-hydroxymethylcytosine, 5-formylcytosine and 5-carboxylcytosine [280,281]. Therefore, the unique DNA methylation profile of a cell is determined by two competing mechanisms: DNA methylation and demethylation. DNA methylation always results from the activity of DNA methyltransferases, whereas DNA demethylation can either be enzymatic or occur via passive loss of the methyl mark during replication in the absence of DNA methyltransferase activity or their low capacity for binding with oxidised forms of 5-methylcytosine [282].

Unlike the myometrium, ULs feature global genome hypomethylation due to decreased expression of de novo DNA methyltransferases *DNMT3A* and *DNMT3B* [283]. By contrast, the expression level of maintenance DNA methyltransferase *DMNT1* in ULs is either higher or equal to that of the myometrium [283,284,285]. A considerable share of hypomethylated genes in ULs is located on chromosome X (*FAM9A*, *CPXCR1*, *CXORF45*, *TAF1*, *NXF5*, *VBP1*, *GABRE*, *DDX53*, *FHL1*, *BRCC3*, *DMD*, *GJB1*, *AP1S2* and *PCDH11X*) [286]. Hypomethylation/expression activation is also typical of collagen genes (*COL4A1*, *COL4A2* and *COL6A3*), prolactin, oestrogen receptor α, repair enzyme gene RAD51B and several oncogenes (*ATP8B4*, *CEMIP*, *ZPMS2*-*AS1*, *RIMS2* and *TFAP2C*) [287,288,289]. Furthermore, the UL has been found to show hypermethylation/inhibited expression of tumour suppressor genes (*EFEMP1*, *FBLN2*, *ARHGAP10*, *HTATIP2*, *DLEC1* and *KRT19*), *RANKL* gene encoding tumour-associated cytokine, glycoprotein Neuregulin 1 *NRG1* gene, 14-3-3γ protein *YWHAG* gene, and transcription factor genes *SATB2*, *KLF4, KLF11* and *ATF3* [119,289,290,291,292,293]. The distribution of aberrantly methylated CpGs in the chromosomes as well as the expression of genes associated with ECM organization, cell adhesion, angiogenesis, endo- and paracrine signalling pathways differ between ULs with and without *MED12* mutations [111] and could also serve as a biomarker for the differential diagnosis of uterine leiomyomas and leiomyosarcomas [292,294].

The DNA methylation patterns described above are more typical of mature tumour cells as the population of stem/progenitor cells in ULs is relatively limited [27,295]. Recent studies have revealed around 10,000 differentially methylated regions, most of which were hypermethylated in UL stem cells in comparison with differentiated ones [295]. In all appearances, the “stem phenotype” of UL cells is determined primarily by the progesterone receptor gene hypermethylation and its inhibited expression. Knocking down this gene in mature UL cells results in a shift of their transcription profile toward non-differentiated cells, decreased ECM production and the activation of genes associated with the cell cycle and proliferation [296]. Hypomethylating agent azacitidine (5-aza-2′-deoxycytidine) activates progesterone receptor expression in UL stem cells and stimulates their differentiation, thus decreasing tumour growth potential [296]. Azacitidine treatment of a primary UL culture decreases tumour cell viability (probably via hypomethylation/activation of tumour-suppressor genes), ECM production and the expression of genes encoding WISP1, c-MYC and MMP7 proteins, which are the targets of the WNT/β-catenin signalling pathway [285].

Global hypomethylation of the UL genome can be also significantly influenced by TET-mediated active DNA demethylation. ULs are characterised by a higher expression of *TET1* and *TET3* than the myometrium [297]. Meanwhile, the expression of *TET1* and *TET3* in UL stem cells is lower than in differentiated ones, in full alignment with the differences in their DNA methylation levels [165]. The *H19*-*let-7*-*TET3* axis is an important expression regulator for the genes associated with the early stages of UL development, signalling pathways and collagen production. In particular, the TET3 protein has the capacity for binding with the promoters of *MED12*, *TGFBR2* (transforming growth factor beta receptor 2) and *TSP1* (thrombospondin 1), thus facilitating their demethylation and increased chromatin availability [298]. *TET1* and *TET3* knockdown or treatment with specific TET inhibitors result in a decrease in both DNA hydroxymethylation levels and UL cell proliferation [297]. *TET3* expression in the UL also depends on sex steroid hormones and increases under combined exposure to oestrogen and progesterone [298]. Furthermore, reduced *TET1* expression in UL cells is associated with the risk of tumour recurrence [299].

Indeed, 5-Hydroxymethylcytosine is the first product of TET-mediated 5-methylcytosine oxidation and is most widely represented in the genome [300]. Located primarily in the transcriptionally active genome regions, 5-Hydroxymethylcytosine levels vary greatly across tissues and ontogenesis stages [301,302,303,304,305,306]. Low 5-hydroxymethylcytosine content is typical of many malignant tumours and is associated with late stages of the disease and poor prognosis [307,308,309]. By contrast, ULs display higher levels of DNA hydroxymethylation than in the adjacent myometrium or leiomyosarcoma [297]. Finally, 5-Hydroxymethylcytosine content does not differ in ULs with and without *MED12* mutations but depends on the menstrual cycle phase: it is higher in the follicular than in the luteal phase [122].

### 6.3. Non-Coding RNAs

Non-coding RNAs (ncRNAs) are involved in transcription, translation, splicing, lengthening of telomeres, genome imprinting, X chromosome inactivation, intercellular communication and more. ULs have been found to feature dysregulation in several RNA classes, namely microRNAs (miRNAs), circular RNAs (circRNAs) and long ncRNAs (lncRNAs).

MiRNAs are short single-stranded RNA molecules containing 18-25 nucleotides. Changes in miRNA profiles in a UL are associated with the expression of components of DNA demethylation, immune response and cell cycle proteins, with the tumour cell differentiation, EDC exposure, tumour size and the patient’s ethnicity [310,311,312,313,314,315]. The most studied miRNAs in UL belong to the *let-7* and *miR-29* families. The *let-7* miRNA is a negative regulator of *HMGA2* mRNA translation, and its expression is reduced in HMGA2-ULs [133]. The *let-7* miRNA may also serve as a marker of cellular senescence and inflammation in ULs. Downregulation of the *miR-29* miRNA is shown to be responsible for TGF-β3- and oestrogen/progesterone-dependent ECM accumulation in UL cells [316]. High *miR-29* content inhibits both the expression of collagen genes and the mechanisms of cell proliferation, invasion and metastasising [317]. Overall, ULs demonstrate expression changes for over 100 miRNAs, often overlapping in their functionality [318].

CircRNAs are stable regulatory RNA molecules covalently closed at the 3′ and 5′ ends. ULs have been found to feature increased expression of 579 and reduced expression of 625 circRNAs as compared to the myometrium [319]. The most significant differences are observed in circRNAs *hsa_circ_0083920*, *hsa_circ_0056686*, *hsa_circ_0062558*, *hsa_circ_0020376* and *hsa_circ_0043597*. In particular, *hsa_circ_0056686* circRNA expression is higher in tumour-associated fibroblasts than in fibroblasts isolated from adjacent myometrial tissues and is associated with UL size, the regulation of proliferation, migration and apoptosis and endoplasmic reticulum stress [319,320].

LncRNAs are RNA molecules containing 200 or more nucleotides. Some variants of the *HOTAIR* long ncRNA gene are associated with the risk of UL development [321]. ULs also display changes in the expression of *H19*, *APTR, XIST, CAR10*, *CASC15*, *TCONS_l2_00000923*, *UC.10*, *HOTTIP*, *LINC00890*, *MEG3*, *LNCRNA-ATB*, *MAMDC2-AS1*, *TSIX*, *HULC*, *BX640708*, *UCA1* and *AK023096* lncRNAs as compared to the myometrium [322]. The *H19* lncRNA is the key regulator of *HMGA2* and *MED12* (“UL drivers”) expression [298]. The *APTR* (Alu-mediated p21 transcriptional regulator) and *XIST* (X-inactive specific transcript) lncRNAs are involved in the hormone-dependent proliferation of UL cells, the activation of the WNT signalling pathway, collagens and fibronectin gene expression [323,324]. Increased expression of the *lnc-AL445665.1-4* lncRNA is typical of multiple but not solitary ULs [325]. The MED12-UL demonstrate changes in the expression of the *SRA1* (steroid receptor RNA activator 1) and *MIAT* (myocardial infarction-associated transcript) lncRNAs, which regulate steroidogenesis and ECM accumulation, respectively [324,326].

### 6.4. Speculations on the Role of Epigenetic Changes in UL Aetiology

Data presented in *6.1-6.3* demonstrate that epigenetic changes in ULs are highly diverse and could be specific either to the condition at large or to patients, molecular subgroups of UL or even cell populations. On the one hand, the epigenetic features of a cell are fairly stable and specific. On the other hand, they are characterised by plasticity and are subject to changes under the influence of endogenous and exogenous stimuli.

Over the last few years, a considerable quantity of experimental and clinical studies have been issued on the role of endocrine-disrupting chemicals (EDCs), in particular, on xenoestrogens (bisphenol A, diethylstilbestrol, phthalates, etc.) in UL development [267,314,327,328,329,330]. Xenoestrogens can mimic the activity of natural hormones and modulate a wide variety of cellular processes (paracrine signalling, proliferation, apoptosis, DNA repair, DNA methylation, histone protein modifications, and gene expression) [331,332,333]. Thus, prenatal and prepubertal exposure to xenoestrogens increases the risk of developing tumours in adulthood [44,334,335,336,337,338,339]. Whereas the specific mechanism of the delayed EDC effect on UL development is yet to be investigated, epigenetic abnormalities in myometrial stem cells are presumed to play a key part in this process [9]. Consequently, the focus of UL aetiology research has somewhat shifted from genetic to epigenetic aberrations.

The search for a universal event that triggers UL development is considerably complicated by the heterogeneity of molecular features across ULs. For a long time, dependence on sex steroid hormones was considered to be the only common trait of all ULs. However, around three decades ago, Bulun et al. demonstrated that over 90% of myomas are also characterised by an increased expression of aromatase, which converts androgens to oestrogens—up to 25 times as high as in the adjacent myometrium. Aromatase expression was also discovered in the myometrium of 75% of women with UL and 0% of women without ULs [340]. A suggestion was made that localised oestrogen biosynthesis may be of pathological significance in the promotion of UL growth [341]. Subsequently, aromatase overexpression was detected in the endometrium and uterine arteries of women diagnosed with UL, which attests to the condition causing systemic lesions [342,343].

Apart from ULs, aromatase overexpression is also observed in other benign uterine neoplasms: endometriosis and adenomyosis [344,345]. Similar to ULs, aromatase expression is found in the eutopic endometrium of endometriosis patients but not in healthy women [346]. In both myomas and endometriomas, aromatase expression is controlled primarily by the proximal promoter II; a few groups of patients have also shown activity of promoters I.3 and I.4 [340,347,348,349,350]. In a benign neoplastic process in the uterus, the potential stimulant of aromatase expression from promoter II is presumed to be prostaglandin E2 [341,347]. Prostaglandin E2 has a wide range of functions, in particular, acting as a smooth muscle tone regulator, inflammation mediator and immune response component. The start of an inflammatory process, prostaglandin E2 production and accumulation may result from harmful environmental exposure [351] and are associated with benign and malignant tumours and dysmenorrhea [352,353,354,355].

Benign neoplasms of the female reproductive tract share a multitude of molecular features that were addressed in detail by Baranov et al. [356]. According to the authors’ hypothesis, dysregulation of mesenchymal stem cells is the source of both endometrial and myometrial neoplasms. However, the fate of a transformed stem cell is determined by its location in the corpus of the uterus and a combination of factors, including genetic and epigenetic background [356]. We suggest that the induction of aromatase expression in uterine stem cells as a result of adverse environmental exposure or a hereditary predisposition may trigger the neoplastic process and be the basis of UL and endometriosis syntropy.

EDC exposure in the early stages of ontogenesis and abnormal aromatase activation may result in a possibility of prepubertal emergence of areas with a high concentration of oestrogen in uterine tissue. Normally, oestrogens trigger the activity of multiple genes and signalling cascades; however, hyperproduction of oestrogens and their oxidised metabolites (catechols and quinones) may also serve as the transforming factor [357,358,359]. The absence of documented pre-menarche UL cases is explained by the fact that, in addition to oestrogen, a progenitor UL cell also needs progesterone, which is produced in the ovaries, to develop proliferative potential [12,360]. Oestrogen activates progesterone receptor expression, and, therefore, regulates the susceptibility of myometrial and UL cells to progesterone [46,361]. As Omar et al. have recently demonstrated, the myometrium of UL patients contains more progesterone receptors and is characterised by higher progesterone-associated gene activation levels [362], which also speaks in favour of systemic but not localised myometrium lesions in UL patients. Exposure to endogenous and exogenous factors may have a cumulative effect, which determines the highest UL risk in premenopausal women. Meanwhile, the observed diversity in morphologic and molecular features of myomas most likely develops at the stage of tumour promotion as a result of sequential epigenetic and genetic abnormalities in the transformed cells (Figure 2).

## 7. Conclusions

UL is a neoplastic disease characterised by a high incidence, diverse clinical presentation, pronounced autoregulation and considerable changeability in the tumour’s genome and epigenome, which surprisingly coincides with a low risk of malignisation. It is generally accepted that a UL develops from a single myometrial cell—presumably a stem cell—as a result of its transformation and clonal expansion. In our opinion, however, the clonal origin of UL once again requires serious investigation in the context of modern data and methodological approaches. The search for an event that triggers UL formation is strongly impeded by the broad range and non-unique nature of predisposing factors, their uneven contribution, on the one hand, and the pronounced variability in clinicopathological, morphological, genetic and epigenetic characteristics of ULs, on the other hand. By the time of their detection and surgical removal, myomas have existed in the uterus for months, years or even decades, undergoing sequential developmental stages and exposure to a variety of endogenous and exogenous factors. As a result, every myoma acquires a unique portrait with a pronounced intercellular heterogeneity, which makes identifying the characteristics of the initially transformed cell highly problematic (Figure 1). Unfortunately, the available techniques do not enable us to establish links across all of the aspects that contribute to the individual profile of each UL cell because focusing on one or several characteristics, we inevitably lose sight of the rest (Figure 1). 

The key role in the neoplastic transformation belongs to genetic and epigenetic alterations, and in this context, the UL is most likely no exception. The wide variety of such aberrations in ULs can be divided into driver events (*MED12* and *FH* mutations, *HMGA2* rearrangements and overexpression, *COL4A5-COL4A6* rearrangements) and passenger events (7q deletions, chromosome 1 rearrangements, heteroploidy, etc.), with the latter occurring in later stages and making the greatest contribution to the intratumoral heterogeneity (Figure 2). Driver mutations in *MED12*, *HMGA2* and other genes appear to be emerging in undifferentiated cells at the early stages of UL tumorigenesis and largely define the fate of each myoma. However, MED12-ULs and HMGA12-ULs have also been demonstrated to feature cells without corresponding driver mutations, which attests to the fact that, instead of emerging in the initial transformed myometrial cell, they appeared later in one of its daughter cells. In light of this, we suggest that the prerequisite of a myometrial cell transformation into a UL cell may be an epigenetic alteration. Furthermore, the prerequisite event and the beginning of UL formation could be separated by a considerable stretch of time. Moreover, we cannot rule out the possibility that the potential prerequisite may occur during gametogenesis (Figure 2). In this case, the transgenerational inheritance of specific epigenetic patterns could be of significance. After puberty, the coinciding of a prerequisite condition with the moment of an imbalance between protective and negative factors results in UL formation. Further on, up until the occurrence of driver mutations, the myoma most likely develops through paracrine interactions with surrounding differentiated myometrial cells. It cannot be ruled out that local changes in hormone concentrations in the myometrium, for instance, caused by aromatase activation, or the emergence of cells with hormonal hypersensitivity could play a protumorigenic role, creating pre-conditioned areas. Therefore, the prerequisite for UL development presumably occurs much earlier than the beginning of tumour formation, while the latter is followed by multiple genetic and epigenetic changes, reflecting the UL evolution.

## Figures and Tables

**Figure 1 ijms-24-05752-f001:**
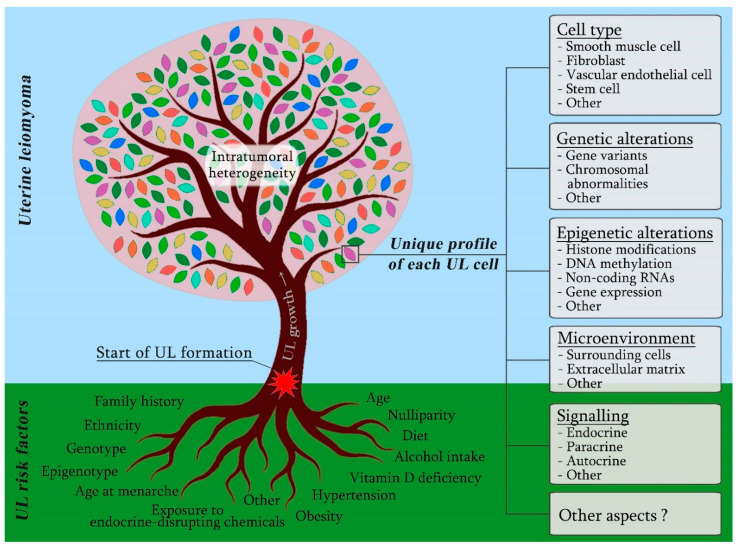
An illustration of predisposing factors intricacy and intratumoral heterogeneity complicating the search for key events of uterine leiomyoma (UL) genesis. UL formation is associated with a wide range of predisposing factors: family medical history, African or Latin American ethnicity, gene variants, epigenetic changes, early menarche, exposure to xenoestrogens, obesity, hypertension, vitamin D deficiency, alcohol intake, dietary habits, nulliparity and advanced reproductive age. During UL growth, each cell acquires a unique set of parameters (cell type, genetic and epigenetic alterations, characteristics of signalling and microenvironmental components), thus resulting in a pronounced intratumoral heterogeneity and complicating identification of the characteristics of initial UL cell(s).

**Figure 2 ijms-24-05752-f002:**
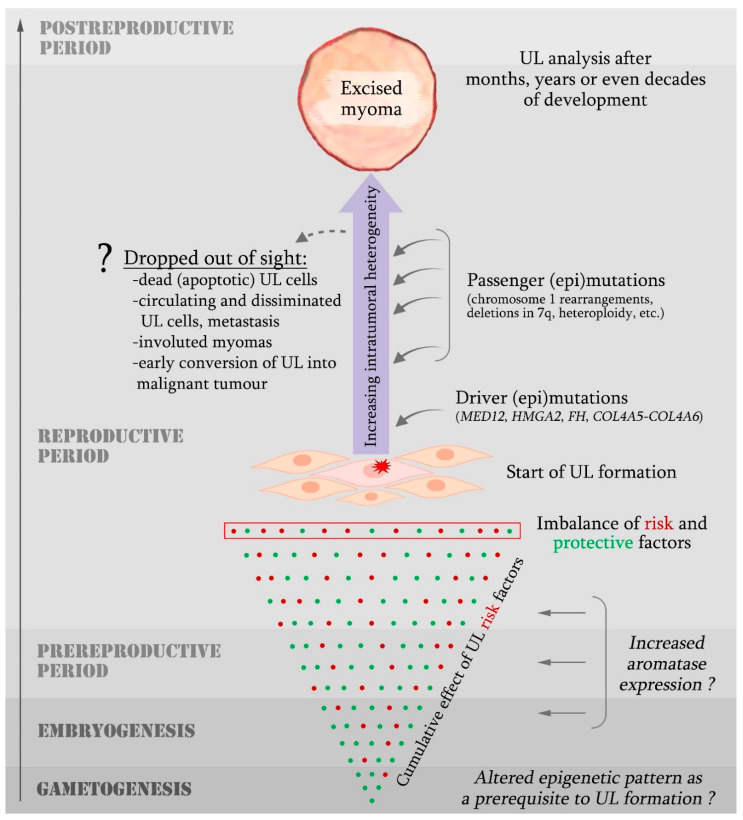
The hypothetic timeline of uterine leiomyoma (UL) genesis. A prerequisite to UL formation may appear in parental gametogenesis and could be presented by altered epigenetic patterns in oocytes or spermatozoa. The accumulation of UL-associated negative effects during ontogenesis can lead to an acute imbalance of risk and protective factors and the beginning of UL formation. UL growth takes a long period of time, during which driver (*MED12* and *FH* mutations, *HMGA2* rearrangements and overexpression, *COL4A5*-*COL4A6* rearrangements) and passenger (7q deletions, chromosome 1 rearrangements, heteroploidy, etc.) changes occur and contribute to the intratumoral heterogeneity.

## Data Availability

Not applicable.

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
