# Peer review of "A View on Uterine Leiomyoma Genesis through the Prism of Genetic, Epigenetic and Cellular Heterogeneityâ€"

_ijms, 2023, doi:10.3390/ijms24065752_

Round 1
Reviewer 1 Report
I have suggested an additional reviewer as the bulk of this paper is outside my area of expertise. Within my own area:
The review does seem to heavily reference other reviews rather than the primary source references which I always find concerning.
In the areas of my own interest; in dicussion hereditary FH syndrome- the authors describe a distinct histological pattern associated with germline FH change- but fail to describe these changes- this would be valuable to enable those reviewing these tumours to understand the triage associated with who should be considered for referral to a genetics unit for testing, particularly given leiomyomas are so common. The authors discuss hereditary inference- but do not go on to provide implications for others in the family and what they might do to lower risk (if anything) to mitigate growth patterns.
Reviewer 2 Report
In this review article, the authors provided a comprehensive overview of the genetic & epigenetic mechanisms of uterine leiomyoma (UL) development. Overall, it is written clearly and is informative. Several minor suggestions are made to further improve the clarity of the article. Please see below.
Figures 1 and 2 are too general and broad and do not capture/summarize the key information discussed in the text. It would be better to provide more specific/informative figures that summarize the text better. For example, the current Fig 1 can be replaced with a new figure summarizing how the interaction between UL cells and the UL microenvironment (e.g. stem cells, fibroblasts, smooth muscle cells, endothelial cells, telocytes) can promote UL development. It would be also helpful to show which cell types express ER & PR. Fig 2 also seems to be too broad. Some items in Fig. 2 were not even discussed in the text (e.g. age, age at menarche, hypertension, alcohol intake as risk factors; oxidative stress, immune response, and inflammation as potential causes). It would be helpful to modify Fig 2 to be more specific so that it can summarize the main text better. In Fig 3, where does ‘increased aromatase expression’ fit? It would be helpful to list some examples of driver vs passenger mutations in Fig 3.
Since section 4 discusses the risk factors, it would be helpful to briefly discuss risk factors shown in Fig 2 such as age, age at menarche, hypertension, and alcohol intake.
Although it was slightly mentioned in the text (lines 421-429), it would be helpful to briefly discuss any existing or developing inhibitors for UL treatment. Is surgery the only treatment for UL? Are there any inhibitors of aromatase, HMGA2, or MED12 being used for UL treatment?
Line 62: Please define the gene name at the first usage (e.g. MED12).
Line 64 says UL are deficient in ER and PR. But lines 80 and 134 indicate UL express ER and PR. Which one is correct?
Line 68: Please provide a few examples of paracrine factors secreted by telocytes or mature cells.
Line 120: What does ‘this pathway’ refer to? Wnt signaling?
Line 145: ‘closely related’ to whom? To women who had UL? Please rephrase the sentence for clarity.
Line 168: What is the small character in front of CELF4?
Line 172: Please specify ‘other’ hereditary syndromes featuring a high risk of UL.
Line 403: Since UL has a lower number of breaks than malignant tumors, it must be 20 or ‘less’?
Line 439: How do benign metastasizing UL metastasize to the lung? Through blood vessels? If benign cells are invasive enough to metastasize to other organs, can they be called ‘benign’? NCI defines benign tumors do NOT invade nearby tissue or spread to other parts of the body.
Line 580: How can decreased EZH2 activity result in increased UL incidence because decreased EZH2 would increase DNA repair gene activity, thereby resulting in less genomic instability and less tumorigenicity?
Line 624: Line 651 says that a hypomethylating agent decreases HL cell viability. But UL feature global genome hypomethylation, which might cause decreased HL cell viability? It is confusing. Please explain.
Line 637: Please briefly explain what differences in DNA methylation profile are between UL with or without MED12 mutations.
Line 692: What is the expression status of mir-29 in UL? Is it decreased?
Line 766: Are uterine stem cells the same as UL stem cells?
Reviewer 3 Report
This article is a comprehensive review of the literature on the etiology, genetic, and epigenetic features of uterine fibroids. The authors have summarized the accumulated literature without bias toward any particular viewpoint. I believe no changes are required.
Reviewer 4 Report
In this review, the authors comprehensively analyze risk and protective factors for uterine leiomyoma (UL) development, UL cellular composition, hormonal and paracrine signaling, epigenetic regulation, and genetic abnormalities.
I think this paper should be published in the journal.
Reviewer 5 Report
The manuscript describes a review of literature in the field of complex pathogenesis of common benign uterine tumours. As I am clinician I can evaluate only a clinical data which I found correct. I believe that from the clinical point of view the manuscript would be highly improved if the authors would explore the relation of pathogenetic details with clinical presentation. Meaning, we know that 70-80 % of women have myomas, however only about one third of them experience symptoms. It is also published that there may be different pathogenetic pathways between women having one single asymptomatic myoma and those having multiple myomas that develop early in life and cause severe symptoms. If 70 to 80 % of all woman have at least one single asymptomatic myoma we may speculate that these tumours may be a natural event occuring inside of uterine myometrium. From clinical point of view and for the future research finding specific target for therapy it would be interesting if authors would present/rewrite the association of pathogenetic pathways with symptomatic multiple myomas. That is why I believe the informative aspect of the manuscript may be improved.
